# TAPEX: TABLE PRE-TRAINING VIA LEARNING A NEURAL SQL EXECUTOR

**Qian Liu[†][∗], Bei Chen[§], Jiaqi Guo[◇][∗], Morteza Ziyadi[♡], Zeqi Lin[§],**
**Weizhu Chen[♡], Jian-Guang Lou[§]**

[†]Beihang University, [◇]Xi'an Jiaotong University, [§]Microsoft Research Asia, [♡]Microsoft Azure AI
qian.liu@buaa.edu.cn, jasperguo2013@stu.xjtu.edu.cn
{bei.chen, morteza.ziyadi, zeqi.lin, wzchen, jlou}@microsoft.com

## ABSTRACT

Recent progress in language model pre-training has achieved a great success via leveraging large-scale unstructured textual data. However, it is still a challenge to apply pre-training on structured tabular data due to the absence of large-scale high-quality tabular data. In this paper, we propose TAPEX to show that table pre-training can be achieved by learning a neural SQL executor over a synthetic corpus, which is obtained by automatically synthesizing executable SQL queries and their execution outputs. TAPEX addresses the data scarcity challenge via guiding the language model to mimic a SQL executor on the diverse, large-scale and high-quality synthetic corpus. We evaluate TAPEX on four benchmark datasets. Experimental results demonstrate that TAPEX outperforms previous table pre-training approaches by a large margin and achieves new state-of-the-art results on all of them. This includes improvements on the weakly-supervised WikiSQL denotation accuracy to $89.5\%$ $(+2.3\%)$, the WikiTableQuestions denotation accuracy to $57.5\%$ $(+4.8\%)$, the SQA denotation accuracy to $74.5\%$ $(+3.5\%)$, and the TabFact accuracy to $84.2\%$ $(+3.2\%)$. To our knowledge, this is the first work to exploit table pre-training via synthetic executable programs and to achieve new state-of-the-art results on various downstream tasks. Our code can be found at https://github.com/microsoft/Table-Pretraining.

## 1 INTRODUCTION

Pre-trained language models (LMs) such as BERT (Devlin et al., 2019) and BART (Lewis et al., 2020) have hit a success on a range of free-form natural language (NL) tasks. By learning from a large amount of unstructured textual data, these models have demonstrated surprising capabilities in understanding NL sentences. Inspired by this huge success, researchers have attempted to extend pre-training to structured tabular data (Herzig et al., 2020; Yin et al., 2020; Yu et al., 2021a; Wang et al., 2021b; Deng et al., 2020; 2021; Shi et al., 2021a). However, different from free-form NL sentences, tabular data often contains rich and meaningful structural information, for which existing pre-training approaches designed for unstructured data are not well suited.

To apply pre-training techniques on structured tabular data, there exist two key challenges: (i) where to obtain a large-scale pre-training corpus with high quality, and (ii) how to design an efficient pre-training task for table pre-training. For the first challenge, existing works generally collect parallel data including NL sentences and tables as the pre-training corpus, since downstream tasks often involve a joint reasoning over both free-form NL sentences and tables. They either crawled tables and their surrounding NL sentences from the Web (Herzig et al., 2020; Yin et al., 2020; Deng et al., 2021), or synthesized NL sentences on available tables (Yu et al., 2021a; Shi et al., 2021a). However, as pointed by Yin et al. (2020), the raw data mined from the Web is extremely noisy and requires complicated heuristics to clean. Conversely, the synthesis method is easier to control the data quality, but it usually requires experts to write hundreds of templates, which is both costly and often lacking diversity. Regarding the pre-training task, existing works often employ different variants of Masked Language Modeling (MLM) (Devlin et al., 2019) to guide LMs to learn better representations of

---

[∗]Work done during an internship at Microsoft Research Asia.

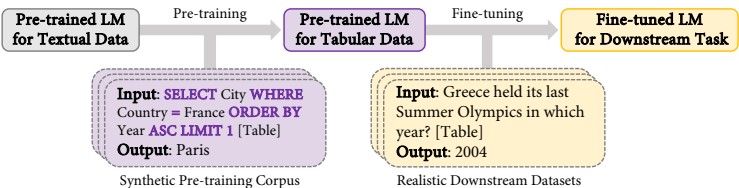

Figure 1: The schematic overview of our method. For the sake of brevity, the table content in the input is simplified with the symbol [Table].

tabular data. For example, TAPAS (Herzig et al., 2020) used MLM with whole word masking, and TABERT (Yin et al., 2020) proposed Masked Column Prediction (MCP) to encourage the model to recover the names and data types of masked columns. Despite their success, they still largely treat tabular data as a structural format of text, which leads to the need of an extremely large corpus for their table pre-training. All of these hinder the progress of table pre-training.

In this paper, we present a novel execution-centric table pre-training approach TAPEX (**TA**ble **Pr**e-training via **EX**ecution). It addresses the above challenges and achieves efficient table pre-training via approximating the structural reasoning process of formal languages over tables. The structural reasoning process is associated with the executability of tables, i.e., tables are inherently capable of supporting various reasoning operations (e.g., summing over a column in the table). In particular, TAPEX approximates the structural reasoning process of SQL queries by pre-training LMs to mimic the behavior of a SQL execution engine on tables. As shown in Figure 1, by sampling executable SQL queries over tables, TAPEX first synthesizes a large-scale pre-training corpus. Then it continues pre-training a language model to output the execution results of these SQL queries, which are obtained from the SQL execution engine. Since the diversity of SQL queries can be systematically guaranteed, we can easily synthesize a diverse, large-scale, and high-quality pre-training corpus. Our key insight is that if a language model can be pre-trained to faithfully "execute" SQL queries and produce correct results, it should have a deep understanding of tables. Thus, the execution pre-training task could be more efficient in understanding tables and reasoning over tables. To our knowledge, TAPEX is the first one to explore table pre-training via synthetic executable programs.

TAPEX is conceptually simple and easy to implement. In this paper, we regard the pre-training as a sequence generation task and employ an encoder-decoder model. Specifically, we employ the pre-trained encoder-decoder language model BART (Lewis et al., 2020) as the backbone. Furthermore, we examine the effectiveness of TAPEX via two fundamental downstream tasks: table-based question answering (TableQA) and table-based fact verification (TableFV). To enable fine-tuning of downstream tasks to take full advantage of TAPEX, we reformulate these tasks using the encoder-decoder sequence generation paradigm. We evaluate TAPEX using four well-known benchmark datasets. Experimental results clearly demonstrate that TAPEX can bring significant and consistent improvements on these datasets. For example, TAPEX obtains an absolute improvement of 19.5% over BART in the WIKITABLEQUESTIONS dataset. Furthermore, TAPEX yields strong results even with a small pre-training corpus, demonstrating its high efficiency. Finally, TAPEX achieves new state-of-the-art results on all experimental benchmarks, outperforming previous approaches by a large margin, including complicated table pre-training approaches with several heuristics in data processing. We will make our code, model, and data publicly available to facilitate future research.

## 2 FINE-TUNING ON DOWNSTREAM TASKS

Before diving into the details of our proposed table pre-training, we start by describing how to tackle downstream task fine-tuning with the encoder-decoder sequence generation paradigm. In this section, we first present the background of two fundamental table related downstream tasks: table-based question answering (TableQA) and table-based fact verification (TableFV). Then we elaborate on our generative fine-tuning method in detail.

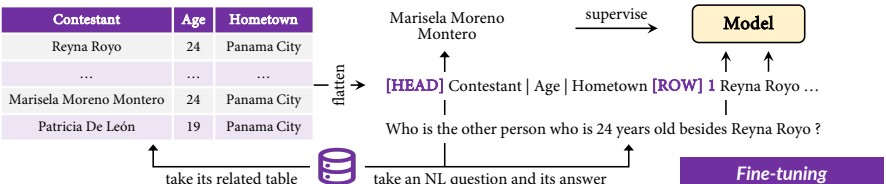

Figure 2: The illustration of the fine-tuning procedure in our method. During fine-tuning, we feed the concatenation of an NL sentence and its corresponding table taken from the downstream task to the model, and train it to output the answer (e.g., "Marisela Moreno Montero").

## 2.1 DOWNSTREAM TASK FORMULATION

As mentioned in § 1, downstream tasks always involve joint reasoning over free-form NL sentences and tables. Therefore, examples of downstream tasks generally contain an NL sentence $\mathbf{x}$ and a (semi-)structured table $T$ as the model input. Each NL sentence consists of $K$ tokens as $\mathbf{x} = x_1, x_2, \cdots, x_K$, while each table $T$ consists of $M$ rows $\{r_i\}_{i=1}^M$, in which each row $r_i$ contains $N$ cell values $\{s_{\langle i,j \rangle}\}_{j=1}^N$. Each cell $s_{\langle i,j \rangle}$ includes a list of tokens and corresponds to a table header $c_j$. As for the output, there are variations among different tasks. In this paper, we focus on TableQA and TableFV. TableQA aims to retrieve table content to answer the user's question, and thus its output is either a list of cell values or number(s) calculated over the selected table region by aggregation functions (e.g., SUM). It is worth noting that for semi-structured tables, the answer may not be exactly table cell values, but their normalized forms (e.g., from 2k to 2,000), which makes downstream tasks more challenging (Oguz et al., 2020). As for TableFV, the output is a binary decision *entailed* or *refused*, indicating whether the NL sentence follows the fact indicated by the table.

## 2.2 GENERATIVE FINE-TUNING

In this section, we present a generative approach for downstream task fine-tuning. Unlike previous works, we model both TableQA and TableFV as sequence generation tasks and leverage generative LMs to generate the output autoregressively. Taking TableQA as an example, given an NL question, our method generates the answer by decoding it in a word-by-word fashion.

**Architecture** Our method theoretically applies for any LM as long as it can generate sequence, such as GPT3 (Brown et al., 2020) and UniLM (Bao et al., 2020). In our experiments, we implemented our method based on BART (Lewis et al., 2020), a widely used pre-trained encoder-decoder model. BART follows a standard sequence-to-sequence Transformer architecture (Vaswani et al., 2017), with modifying ReLU activation functions to GeLU. It is pre-trained via corrupting sentences (i.e., randomly sampling length-variable spans and masking each one with a single [MASK] token) and then optimizing a reconstruction loss. As for the number of layers, we employ the BART$_{\text{Large}}$ configuration in our experiments, i.e., 12 layers are used in both the encoder and the decoder.

**Model Input** As illustrated in Figure 2, the input contains an NL sentence and its corresponding table. Encoding the NL sentence is relatively straightforward, while encoding the table is non-trivial since it exhibits underlying structures. In practice, we flatten the table into a sequence so that it can be fed directly into the model. By inserting several special tokens to indicate the table boundaries, a flattened table can be represented as $T^* = [\text{HEAD}], c_1, \cdots, c_N, [\text{ROW}], 1, r_1, [\text{ROW}], 2, r_2, \cdots, r_M$. Here [HEAD] and [ROW] are special tokens indicating the region of table headers and rows respectively, and the number after [ROW] is used to indicate the row index. Notably, we also separate headers or cells in different columns using a vertical bar $|$. Finally, we prefix the flattened table $T^*$ with the NL sentence $\mathbf{x}$ and feed them into the model encoder.

**Model Output** With attending on the encoder, the decoder is responsible for modeling the outputs of both TableQA and TableFV. For TableQA, the output is the concatenation of the answer(s) separated by commas, and the decoder generates it autoregressively. In this way, our model can readily support (almost) all operators and their compositions in TableQA. For TableFV, as BART does for sequence classification tasks (Lewis et al., 2020), the same input is fed into both the encoder and decoder, and a binary classifier upon the hidden state of the last token in the decoder is used for the output. Notably, our method can be easily extended to other table related tasks in a similar way.

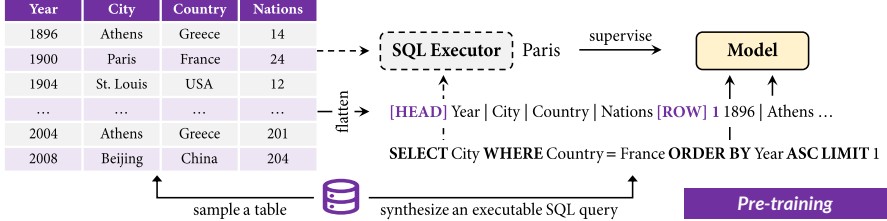

Figure 3: The illustration of the pre-training procedure in our method. During pre-training, we feed the concatenation of a sampled SQL query and a sampled table to the model, and train it to output the corresponding execution result (e.g., "Pairs").

**Fine-Tuning Strategy** Since our approach can perform various downstream tasks on the same architecture, it can easily perform multi-task learning. Therefore, we explore two ways of fine-tuning, one for vanilla fine-tuning and the other for multi-task fine-tuning. The former is to fine-tune the model on each individual downstream task. The latter is inspired by TAPAS (Herzig et al., 2020) and T5 (Raffel et al., 2020), which first fine-tunes the model on related or similar intermediate downstream tasks and then continues to fine-tune it on the target downstream task.

**Discussion** Our approach comes with several advantages: (i) *Flexibility*: due to the powerful expressiveness of encoder-decoder models, our approach can readily adapt to (almost) any kind of output. (ii) *Conveniency*: our approach does not require any modification (e.g., table-specific masking) on pre-trained LMs, and can be trained in an end-to-end manner. (iii) *Transferability*: since we formulate downstream tasks as sequence generation tasks, which allows different tasks to share the same training protocol, it is easy to perform multi-task fine-tuning for our approach.

## 3 TABLE PRE-TRAINING VIA EXECUTION

As mentioned in § 1, TAPEX achieves efficient table pre-training by training LMs to mimic the behavior of a SQL execution engine. In this section, we illustrate how to conduct table pre-training from two aspects: the pre-training task and the pre-training corpus.

### 3.1 PRE-TRAINING TASK

Following the MLM task in NL pre-training, existing works usually use reconstruction tasks for table pre-training. They generally take corrupted tables and NL sentences as input and try to recover the corrupted parts, in order to strengthen the linking between NL sentences and tables. While these pre-training tasks perform well, they tend to be less efficient since they usually require an extremely large pre-training corpus.

To design efficient tasks for table pre-training, we argue that the key lies in the executability of tables. That is to say, structured tables enable us to perform discrete operations on them via programming languages such as SQL queries, while unstructured text does not. Taking this into account, TAPEX adopts SQL execution as the **only** pre-training task. As illustrated in Figure 3, the pre-training of TAPEX is similar to the procedure of the above generative fine-tuning. Given an executable SQL query and a table $T$, TAPEX first concatenates the SQL query and the flattened table $T^*$ to feed into the model encoder. Then it obtains the query's execution result through an off-the-shelf SQL executor (e.g., MySQL) to serve as the supervision for the model decoder. Intuitively, the pre-training procedure is to encourage a language model to be a neural SQL executor. We believe that if a language model can be trained to faithfully "execute" SQL queries and produce correct results, then it should have a deep understanding of tables.

### 3.2 PRE-TRAINING CORPUS

Synthesizing the pre-training corpus is very important for table pre-training. Generally, there are two key factors: the table source and the SQL query sampling strategy.

| Model | Dev | Test |
|---|---|---|
| *Previous Systems* | | |
| Guo & Gao (2019) | 61.1 | 61.0 |
| Liang et al. (2018) | 71.8 | 72.4 |
| Agarwal et al. (2019) | 74.9 | 74.8 |
| Wang et al. (2019b) | 79.4 | 79.3 |
| *Pre-trained Language Models* | | |
| Min et al. (2019) | 84.4 | 83.9 |
| *w.* Execution-Guided Decoding | 87.4 | 87.2 |
| Herzig et al. (2020) | 85.1 | 83.6 |
| Yu et al. (2021a) | 85.9 | 84.7 |
| BART | 87.3 | 85.8 |
| TAPEX | **89.2** | **89.5** |

Table 1: Denotation accuracies on WIKISQL-WEAK. *Execution-Guided Decoding* is proposed to leverage execution results of SQL queries during inference (Wang et al., 2018).

| Model | Dev | Test |
|---|---|---|
| *Previous Systems* | | |
| Pasupat & Liang (2015) | 37.0 | 37.1 |
| Neelakantan et al. (2016) | 34.1 | 34.2 |
| Zhang et al. (2017) | 40.6 | 43.7 |
| Liang et al. (2018) | 42.7 | 43.8 |
| Dasigi et al. (2019) | 43.1 | 44.3 |
| Agarwal et al. (2019) | 43.2 | 44.1 |
| Wang et al. (2019b) | 43.7 | 44.5 |
| *Pre-trained Language Models* | | |
| Herzig et al. (2020) | – | 48.8 |
| Yin et al. (2020) | 53.0 | 52.3 |
| Yu et al. (2021a) | 51.9 | 52.7 |
| BART | 37.2 | 38.0 |
| TAPEX | **57.0** | **57.5** |

Table 2: Denotation accuracies on WIKITABLEQUESTIONS.

**Table Source** Following previous work by Yin et al. (2020), we choose publicly available semi-structured tables as the table source. However, rather than requiring millions of raw tables in (Yin et al., 2020), TAPEX works well even with only a few thousand tables. Therefore, instead of fetching noisy tables from the Web and then heuristically filtering them, we pick high-quality tables right from existing public datasets. Concretely, we randomly select nearly $1,500$ tables from the training set of WIKITABLEQUESTIONS (Pasupat & Liang, 2015) as the table source for our pre-training corpus. Notice that there is no overlap between the tables used in our pre-training and the tables used in the dev and test sets of all downstream tasks, so there is no data leakage problem.

**Query Sampling** Regarding the sampling of diverse SQL queries, there are various choices in the literature. We can either sample SQL queries according to a probabilistic context-free grammar (Wang et al., 2021a), or instantiate SQL templates over different tables (Zhong et al., 2020a). In our experiments, we follow the latter, where SQL templates are automatically extracted from the SQUALL dataset (Shi et al., 2020b). An example SQL template is: SELECT $num_1$ WHERE $text_1$ = $val_1$, where $num_1$ and $text_1$ correspond to a numeric column and a text column respectively, and $val_1$ refers to one of the cell values with respect to the column $text_1$. Given a SQL template, at each instantiation, we uniformly sample headers and cell values from a sampled table to fill the template, forming a concrete SQL query. Notably, SQL queries that execute with empty results are discarded, because empty results do not reflect much information about the executability of tables. This way, we can obtain a large-scale pre-training corpus with high quality.

## 4 EXPERIMENTS

In this section, we evaluate TAPEX on different downstream tasks to verify its effectiveness.

**Dataset and Evaluation** We evaluate the performance of our approach on weakly-supervised WikiSQL (WIKISQL-WEAK) (Zhong et al., 2017), WIKITABLEQUESTIONS (Pasupat & Liang, 2015), SQA (Iyyer et al., 2017), and TABFACT (Chen et al., 2020). Compared to WIKISQL-WEAK, which only requires filtering and optionally aggregating on table cell values, WIKITABLEQUESTIONS requires more complicated reasoning capabilities. SQA is a conversational benchmark, which requires our approach to model the conversational context. Datset details can be found in Appendix A. For TableQA datasets, the evaluation metric is denotation accuracy, which checks whether the predicted answer(s) is equal to the ground-truth answer(s). It is worth noting that we evaluate our approach on WIKISQL-WEAK with answer annotations provided by TAPAS (Herzig et al., 2020), since nearly $2\%$ of answers obtained from the official evaluation script are incorrect. For TABFACT, the evaluation metric is accuracy, which is calculated using the percentage of correct prediction.

**Implementation Details** We implement our approach based on fairseq (Ott et al., 2019). During pre-training, we synthesize up to $5$ million pairs of SQL queries and their execution results for

| Model | ALL | SEQ | $Q_1$ | $Q_2$ | $Q_3$ |
|---|---|---|---|---|---|
| *Previous Systems* | | | | | |
| Pasupat & Liang (2015) | 33.2 | 7.7 | 51.4 | 22.2 | 22.3 |
| Neelakantan et al. (2017) | 40.2 | 11.8 | 60.0 | 35.9 | 25.5 |
| Iyyer et al. (2017) | 44.7 | 12.8 | 70.4 | 41.1 | 23.6 |
| Liu et al. (2019) | – | – | 70.9 | 39.5 | – |
| Sun et al. (2019) | 45.6 | 13.2 | 70.3 | 42.6 | 24.8 |
| Mueller et al. (2019) | 55.1 | 28.1 | 67.2 | 52.7 | 46.8 |
| *Pre-trained Language Models* | | | | | |
| Yu et al. (2021b) | 65.4 | 38.5 | 78.4 | 65.3 | 55.1 |
| Herzig et al. (2020) | 67.2 | 40.4 | 78.2 | 66.0 | 59.7 |
| Eisenschlos et al. (2020) | 71.0 | 44.8 | **80.9** | 70.6 | 64.0 |
| BART | 58.6 | 27.8 | 65.3 | 54.1 | 57.0 |
| TAPEX | **74.5** | **48.4** | 76.2 | **71.9** | **76.9** |

Table 3: Denotation accuracies on SQA test set. ALL is the denotation accuracy over all sentences, SEQ the denotation accuracy over all conversations, and $Q_i$ the denotation accuracy of the i-th sentence in a conversation.

| Model | Dev | Test | Test$_{simple}$ | Test$_{complex}$ | Test$_{small}$ |
|---|---|---|---|---|---|
| *Pre-trained Language Models* | | | | | |
| Chen et al. (2020) | 66.1 | 65.1 | 79.1 | 58.2 | 68.1 |
| Zhong et al. (2020b) | 71.8 | 71.7 | 85.4 | 65.1 | 74.3 |
| Shi et al. (2020a) | 72.5 | 72.3 | 85.9 | 65.7 | 74.2 |
| Zhang et al. (2020) | 73.3 | 73.2 | 85.5 | 67.2 | – |
| Yang et al. (2020) | 74.9 | 74.4 | 88.3 | 67.6 | 76.2 |
| Eisenschlos et al. (2020) | 81.0 | 81.0 | 92.3 | 75.6 | 83.9 |
| BART | 81.2 | 80.8 | 90.7 | 76.0 | 82.5 |
| TAPEX | **84.6** | **84.2** | **93.9** | **79.6** | **85.9** |
| Human Performance | - | - | - | - | 92.1 |

Table 4: Accuracies on TABFACT, including the Human Performance.

TAPEX. In the following, unless specified explicitly, all the experimental results are by default evaluated under the 5 million setting. Our pre-training procedure runs up to $50,000$ steps with a batch size of 256. It takes about 36 hours on 8 Tesla V100 GPUs to finish the pre-training. The best pre-training checkpoint is selected based on the loss on the validation set. For all downstream datasets, the fine-tuning procedure runs up to $20,000$ steps with a batch size of 128. For both pre-training and fine-tuning, the learning rate is $3 \times 10^{-5}$.

## 4.1 MAIN RESULTS

Table 1, Table 2, Table 3 and Table 4 summarize the experimental results of various models on WIKISQL-WEAK, WIKITABLEQUESTIONS, SQA and TABFACT respectively. For both dev and test sets of all datasets, we report the median performance of our approach for five random runs.

**WIKISQL-WEAK** As shown in Table 1, TAPEX outperforms all the baselines by a large margin. On the test set of WIKISQL-WEAK, TAPEX registers a denotation accuracy of $89.5\%$, which is $3.7\%$ higher than BART and $2.3\%$ higher than the previous best performance. This is significant since the previous best model has already utilized the execution-guided decoding. In short, TAPEX achieves a new state-of-the-art result on the well-known benchmark WIKISQL-WEAK.

**WIKITABLEQUESTIONS** On the more challenging WIKITABLEQUESTIONS, TAPEX also achieves a new state-of-the-art denotation accuracy of $57.5\%$, surpassing the previous best system by $4.8\%$ (Table 2). Meanwhile, we find that BART alone can only reach the denotation accuracy of $38.0\%$, much worse than the performance of previous pre-training models. We conjecture that the performance degradation could be attributed to the relatively small amount of training data in WIK-ITABLEQUESTIONS, which makes the adaptation of BART to tabular structures more challenging.

> Who are the only players listed that played in 2011 ? [HEAD] player | year | round | result |
> opponent [ROW] 1 ray mond van bar ne ve ld | 2009 | quarter - final | won | j elle k la as en
> [ROW] 2 ray mond van bar ne ve ld | 2010 | 2 nd round | won | bre nd an d olan [ROW] 3 ad
> rian le w is | 2011 | final | won | g ary and erson

Figure 4: The visualization results of attention weights from other tokens to the cell "adrian lewis". Intuitively, the darker the color, the more closely the word is associated with "adrian lewis".

However, TAPEX delivers a dramatic improvement of 19.5% over BART, indicating that in the low data regime, the improvements introduced by TAPEX are often more significant.

**SQA** Table 3 presents the performance of various models on the test set of SQA, where TAPEX again obtains a new state-of-the-art denotation accuracy in terms of both the conversation level (48.4%) and the sentence level (74.5%). This improvement is also a surprise to us since SQA is a conversational dataset while our pre-training task is context-free. Meanwhile, the substantial improvements of TAPEX over BART on SQA continues to verify the same observation that TAPEX alleviates the low resource issue.

**TABFACT** Beyond TableQA, TAPEX also excels at TableFV. As shown in Table 4, TAPEX achieves new state-of-the-art results on all subsets of TABFACT. For example, it surpasses the previous best system by 4.0% on $\text{Test}_{\text{complex}}$. The result shows that TAPEX endows BART with generic table understanding capabilities, which could be adapted to different downstream tasks, regardless of whether these tasks are highly similar to the TAPEX pre-training task or not.

**Overall Results** Experimental results on four datasets show that TAPEX can broadly improve the model ability on understanding tables, especially in the low data regime.

### 4.2 MULTI-TASK RESULTS

As discussed in § 2.2, our approach can easily perform multi-task learning, thereby conferring benefits to downstream tasks. To verify it, we conducted multi-task fine-tuning experiments and obtained the following findings: (1) when initialized by BART, multi-task fine-tuning boosts the performance of the target task significantly; (2) when initialized by TAPEX, the gain of multi-task fine-tuning tends to be marginal, suggesting that most of the "skills" (loosely speaking) gained by multi-task learning can be acquired by our table pre-training. Detailed results can be found in Appendix B.

## 5 ANALYSIS

In this section, we carefully analyze our approach in terms of various aspects. Besides, we perform an exploratory analysis to provide more insights for future work, which can be found in Appendix C.

**SQL Execution by Pre-training** In order to understand how well TAPEX performs SQL execution after pre-training, we analyze its performance on nearly 20,000 held-out SQL queries over unseen tables. Overall, the SQL execution accuracy is relatively high, as TAPEX correctly "executes" 89.6% of the SQL queries[1]. In particular, TAPEX performs better on `Filter`, `Aggregate` and `Superlative` operators, indicating that it is highly accurate in table cell selection and table aggregating. Regarding `Arithmetic` and `Comparative` operators, TAPEX also does a good job, demonstrating its numerical reasoning skill on tables. To summarize, TAPEX has learned to be a neural SQL executor with good selection, aggregating and numerical capabilities.

**Table Understanding by Pre-training** To provide insight on if TAPEX helps downstream tasks understand tables better, we visualize and analyze the self-attention of TAPEX (without fine-tuning) on sampled WIKITABLEQUESTIONS examples. As shown in Figure 4, TAPEX seems to focus more on the row and the header where a cell corresponds to. Taking the example from Figure 4, the attention weights imply that "adrian lewis" is closely associated with the first column "player" and the entire third row, which are the positions of "adrian lewis" in the structured table.

**Table Reasoning by Pre-training** To understand if TAPEX can improve table reasoning, we compare the performance of TAPEX to BART on 500 randomly selected questions and manually ana-

---

[1]The full analysis about SQL execution can be found in Appendix D.

| Operator | Example Question | BART | TAPEX |
|---|---|---|---|
| **Select** | What is **the years won** for each team? | 41.3% | 64.8% (+23.5%) |
| **Filter** | How long did **Taiki Tsuchiya** last? | 40.1% | 65.7% (+25.6%) |
| **Aggregate** | What is the **amount of** matches drawn? | 26.9 % | 57.4% (+30.5%) |
| **Superlative** | What was the **last** Baekje Temple? | 46.3 % | 64.3% (+18.0%) |
| **Arithmetic** | What is the **difference** between White voters and Black voters in 1948? | 33.1 % | 53.5% (+20.4%) |
| **Comparative** | Besides Tiger Woods, what other player won **between 2007 and 2009**? | 30.0 % | 55.9% (+25.9%) |
| **Group** | What was score **for each** winning game? | 49.5 % | 66.7% (+17.2%) |

Table 5: The most common operators in the randomly selected 500 questions from WIKITABLE-QUESTIONS dev set. Listed are, the operator, the example question with the operator semantic (i.e., the **colorful** spans), the performance of BART and TAPEX on the operator.

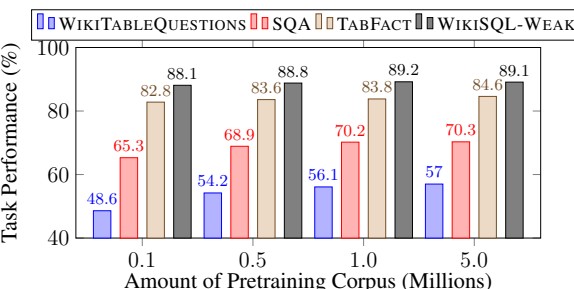

Figure 5: The illustration of downstream tasks performance with different scales of pre-training corpus. Scaling up the pre-training corpus of TAPEX generally brings positive effects across datasets.

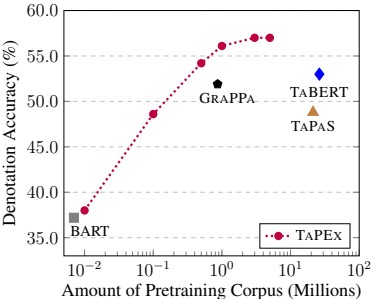

Figure 6: The amount of pre-training corpus vs. denotation accuracy on WIKITABLE-QUESTIONS dev set. TAPEX surpasses existing table pre-training approaches with a much smaller corpus, showing its high efficiency.

lyzed them in Table 5. One can find that TAPEX significantly boosts the performance on all operators, implying that it does enhance BART's capabilities for joint reasoning over text and tables.

**The Scale of Pre-training Corpus**  Figure 5 illustrates downstream performance with different scales of the pre-training corpus. It can be seen that even if our pre-training corpus is synthetic, scaling up the pre-training corpus generally brings positive effects. The observation is analogous to the one in language modeling (Brown et al., 2020): the larger the pre-training corpus, the better the downstream performance. By the comparison across different datasets, we can find that for simple tasks like WIKISQL-WEAK, the gains by scaling up pre-training corpus become marginal, while they remain non-trivial for complex tasks like TABFACT. Meanwhile, both downstream datasets in the low data regime show a positive trend by increasing the pre-training corpus. Conclusively, the scale matters when the downstream task is difficult, or the downstream dataset is relatively small.

**The Efficiency of Pre-training**  As mentioned in § 1, the pre-training efficiency of existing table pre-training approaches is relatively low, as they usually require an extremely large corpus. Therefore, taking WIKITABLEQUESTIONS as an example, we compare the pre-training efficiency of TAPEX with TAPAS (Herzig et al., 2020), TABERT (Yin et al., 2020) and GRAPPA (Yu et al., 2021a). It is worth noting that part of the pre-training corpus for GRAPPA comes from human-annotated, high-quality parallel data. As shown in Figure 6, TAPEX can yield very promising performance when using a much smaller pre-training corpus, indicating that our proposed SQL execution pre-training task is more efficient than other table pre-training tasks.

**Limitations**  The first limitation of our approach is that it cannot ideally handle large tables. As mentioned above, we employ the table flattening technique to represent a table. It works well when the table is relatively small, but it becomes infeasible when the table is too large to fit in memory. In practice, we can compress tables by removing some unrelated rows or columns, which would decrease downstream performance. The second limitation is that the task of text-to-SQL cannot

benefit from our proposed table pre-training. We have tried to apply TAPEX for a text-to-SQL task, where the input remains the same and the output converts to SQL. However, TAPEX does not show a significant advantage over BART. We attribute this to two factors: first, our synthetic pre-training corpus does not contribute to grounding, one of the most important factors for semantic parsing (Liu et al., 2021); second, table reasoning capabilities (e.g., aggregate) learned by TAPEX may not be necessary for SQL generation. For example, a model could still understand an NL phrase "total" as the aggregation function "sum", even though it is unaware of the mathematical meaning of "sum".

## 6 RELATED WORK

**Table Pre-training**   The work most related to ours is table pre-training whose key factors include the pre-training corpus and the pre-training task. As for the pre-training corpus, most of previous works almost collect NL-table data to perform table pre-training. They either mined a large corpus of tables and their NL sentence contexts (Yin et al., 2020; Herzig et al., 2020), leveraged human-annotated parallel NL-table datasets for pre-training (Deng et al., 2021; Yu et al., 2021a), or synthesized a NL-table corpus using human-written templates (Yu et al., 2021a; Eisenschlos et al., 2020). Our work is different from theirs because we are the first to use pure synthetic SQL-table data for table pre-training, which allows us to automatically synthesize a diverse, large-scale, and high-quality pre-training corpus. As for the pre-training task, existing works proposed several pre-training tasks, such as Mask Column Prediction (Yin et al., 2020), Multi-choice Cloze at the Cell Level (Wang et al., 2021b) and Structure Grounding (Deng et al., 2021). Different from all of them, we present a novel SQL execution task to perform table pre-training.

**Joint Understanding on Table and Text**   As our experiments are mainly on TableQA and TableFV, our work is also closely related to previous methods for these tasks. For TableQA, previous works almost formulate it as a weakly semantic parsing task (Liang et al., 2018; Wang et al., 2019a; Guo et al., 2021), which always employ reinforcement learning to optimize semantic parsers over tables. Although these parsers produce logic forms (e.g., SQL), they have difficulties in training due to the large search space and the presence of spurious programs (Goldman et al., 2018). In addition, another promising line of work has emerged in recent advances (Mueller et al., 2019; Herzig et al., 2020), which aims at answering NL sentences without logical forms. This line of work predicts answer(s) by selecting cell values and optionally applying an aggregation operator to them. They can be easily trained, but their modeling ability is limited. For example, it is hard to support compound aggregation operators such as `max(Year) - min(Year)`. What makes our approach different from these works is that we employ generative models to handle TableQA and can enjoy the end-to-end training and flexibility simultaneously. For TableFV, previous works usually employ specialized architectures with limited scalability (Shi et al., 2020a; Yang et al., 2020; Shi et al., 2021b). For example, Zhong et al. (2020b) leveraged a graph construction mechanism, a semantic parser, and a semantic composition model to capture the connections among the NL sentence and the table. While the approach works well for TableFV, it is not easily applied to other table-related tasks. Compared with them, our approach works well for a variety of downstream tasks in the same architecture.

## 7 CONCLUSION

In this paper, we present TAPEX, an execution-centric table pre-training approach whose corpus is automatically synthesized via sampling SQL queries and their execution results. TAPEX addresses the data scarcity challenge in table pre-training by learning a neural SQL executor on a diverse, large-scale, and high-quality synthetic corpus. Experimental results on four downstream datasets demonstrate that TAPEX outperforms previous table pre-training approaches by a large margin and achieves new state-of-the-art results on all of them. Our work opens the way to exploit structured data by pre-training on synthetic executable programs, which is conceptually simple and has great potential to be extended to other research areas (e.g., knowledge base).

## ACKNOWLEDGEMENT

We would like to thank all the anonymous reviewers for their constructive feedback. The first author Qian is supported by the Academic Excellence Foundation of Beihang University for PhD Students.

## ETHICS STATEMENT

In this work, we present a novel pre-training approach for tabular data, which approximates the structural reasoning process of formal languages over tables to achieve efficient table pre-training. Different from previous works which employ web crawling to construct a large-scale NL-table corpus for pre-training, our pre-training corpus is synthesized via sampling SQL queries and their execution results on public tables. Compared with previous works, our pre-training corpus is more controllable with high-quality. For example, compared with TABERT which crawls 26 million noisy tables from the Web, our approach adopts $1,500$ high-quality tables from public datasets, which greatly alleviates the potential privacy and bias issues raised by web crawling. We evaluate our approach on two fundamental table-related tasks: table-based question answering and table-based fact verification. The former enables non-expert users to query databases without learning programming languages, while the latter helps users to verify whether a textual hypothesis is valid based on given tabular evidence. Experimental results on four well-known benchmark datasets show that our approach achieves new state-of-the-art results on all of them, especially in the low data regime.

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

| Task | Dataset | Type | # Sentences | # Tables |
|------|---------|------|-------------|----------|
| TableQA | WIKISQL-WEAK | Simple QA | 80, 654 | 24, 241 |
| | WIKITABLEQUESTIONS | Complex QA | 22, 033 | 2, 108 |
| | SQA | Conversation QA | 17, 553 | 982 |
| TableFV | TABFACT | Fact Verification | 118, 275 | 16, 573 |

Table 6: Experimental dataset statistics.

| Dataset | Example Input | Example Output |
|---------|---------------|----------------|
| WIKISQL-WEAK | How many CFL teams are from York College? [HEAD] : pick # | CFL team Player | Position | College [ROW] 1 : 27 | hamilton tiger-cats | connor healey | db | wilfrid laurier [ROW] 2 : 28 | calgary stampeders | anthony forgione | ol | york . . . | 2 |
| WIKITABLEQUESTIONS | Which album released by the band schnell fenster produced the most singles appearing on the australian peak chart? [HEAD] : Year | Title | Peak Chart Positions AUS | Peak Chart Positions NZ | Album [ROW] 1 : 1988 | "whisper" | 58 | 42 | the sound of trees [ROW] 2 : 1988 | "love-hate relationship" | 81 | 46 | The Sound Of Trees . . . | The Sound Of Trees |
| SQA | where are the players from? which player went to louisiana state university? [HEAD] : Pick | Player | Team | Position | School [ROW] 1 : 1 | Ben McDonald | Baltimore Orioles | RHP | Louisiana State University [ROW] 2 : Tyler Houston | Atlanta Braves | C | Valley HS (Las Vegas, NV) . . . | Ben McDonald |
| TABFACT | On june 26th, 2010 kyle busch drove a total of 211.6 miles at an average speed of 110.673 miles per hour. [HEAD] : year | date | driver | team | manufacturer | laps | - | race time | average speed (mph) [ROW] 1 : 1990 | july 15 | tommy ellis | john jackson | buick | 300 | 317.4 (510.805) | 3:41:58 | 85.797 [ROW] 2 : 1990 | october 14 | rick mast | ag dillard motorsports | buick | 250 | 264.5 (425.671) | 2:44:37 | 94.45 . . . | 1 (Yes) |

Table 7: The example inputs and outputs for our model on experimental datasets.

# A  DOWNSTREAM DATASETS

The dataset statistics are shown in Table 6, while Table 7 show example inputs and outputs for our model. Note that SQA is a conversation benchmark, and we directly concatenate the history and $i$-th question as the "sentence" part ($\mathbf{x}$) in the input, as done in Liu et al. (2020).

# B  MULTI-TASK RESULTS

Table 8 presents the full experimental results on multi-task fine-tuning mentioned in § 2.2. Note that we chose WIKISQL-WEAK and TABFACT as the transfer source because their training data are relatively rich.

# C  EXPLORATORY ANALYSIS

In this section, we perform an exploratory analysis to provide more insights for future work. Concretely, we explore two interesting research questions: (1) How does the difficulty of SQL queries

| Source ↦ Target | BART | TAPEX |
|---|---|---|
| TABFACT ↦ WIKITABLEQUESTIONS | 42.5 | 58.5 |
| WIKISQL-WEAK ↦ WIKITABLEQUESTIONS | 47.4 | 57.2 |
| WIKITABLEQUESTIONS | 37.2 | 57.0 |
| TABFACT ↦ SQA | 62.1 | 71.0 |
| WIKISQL-WEAK ↦ SQA | 64.1 | 70.8 |
| SQA | 57.5 | 70.3 |

Table 8: Experimental results (denotation accuracy) of multi-task fine-tuning on the **Target** dev set. **Source** ↦ **Target** means first fine-tuning on **Source** and then fine-tuning on **Target**.

| Difficulty | Example SQL Query |
|---|---|
| Easy | **SELECT** Date
**SELECT COUNT** (Canal)
**SELECT** Name **WHERE** Age **>=** 28 |
| Medium | **SELECT** Region **ORDER BY** ID **DESC LIMIT 1**
**SELECT COUNT** (Tornadoes) **WHERE** Date **=** 1965
**SELECT** District **WHERE** District **!=** "Tikamgarh" **AND** Agg **=** 0 |
| Hard | **SELECT** (**SELECT COUNT**( Distinct Area)) **>=** 5
**SELECT COUNT** (*) **WHERE** Result **=** "won" **AND** Year **>** 1987
**SELECT** Driver **WHERE** Manufacturer **=** "t-bird" **ORDER BY** Pos **ASC LIMIT 1** |
| Extra Hard | **SELECT COUNT** (*) **WHERE** Position **=** 1 **AND** Notes **=** "110 m hurdles" **AND** Year **>** 2008
**SELECT** Nation **WHERE** Nation **!=** "Japan" **AND** Gold **=** (**SELECT** Gold **WHERE** Nation **=** "Japan" )
**SELECT** Tournament **WHERE** Tournament **IN** ("oldsmar", "los angeles") **GROUP BY** Tournament **ORDER BY COUNT** (*) **DESC LIMIT 1** |

Table 9: Four SQL query difficulty levels and their corresponding example SQL queries.

in pre-training impact the performance of downstream tasks? (2) Would it be better to use natural language sentences instead of SQL queries during pre-training?

## C.1 IMPACT OF SQL QUERY DIFFICULTY IN PRE-TRAINING

**SQL Difficulty Criteria** Inspired by Yu et al. (2018), we suppose that the difficulty of a SQL query can be measured by the number of SQL elements. An element can be either a SQL keyword (e.g., **SELECT**), or a table schema (i.e., a header or a cell value). In practice, we obtain elements of SQL queries via an off-the-shelf SQL parser [2], which returns a stream of SQL elements for each SQL query. Empirically, we categorize SQL queries with $\leq 6$ elements into *Easy*, $> 6$ and $\leq 14$ elements into *Medium*, $> 14$ and $\leq 20$ elements into *Hard*, and the rest into *Extra Hard*. Example SQL queries of different difficulty levels can be found in Table 9. Based on the SQL difficulty criteria, we divide the templates from SQUALL (Shi et al., 2020b) into four levels of difficulty and gradually add them to the construction of the pre-training corpus from Easy-level ($\leq$ Easy) to Extra-Hard-level ($\leq$ Extra Hard). Notably, to avoid the effect of the scale of pre-training, we maintain the same amount of examples for the above pre-training corpus.

**Downstream Performance** The experimental results are shown in Figure 7. As can be seen, it is helpful to add harder SQL queries to the pre-training corpus in most cases. For example, compared to $\leq$ Easy, $\leq$ Medium achieves consistent improvements on the performance of downstream tasks (e.g., 10.6% on WIKITABLEQUESTIONS). Meanwhile, we also notice that the impact of the difficulty of SQL queries becomes less significant after the Medium-level. On the TABFACT dataset, involving Extra-Hard-level SQL queries in pre-training even slightly hurts the performance.

---

[2] https://github.com/forward/sql-parser

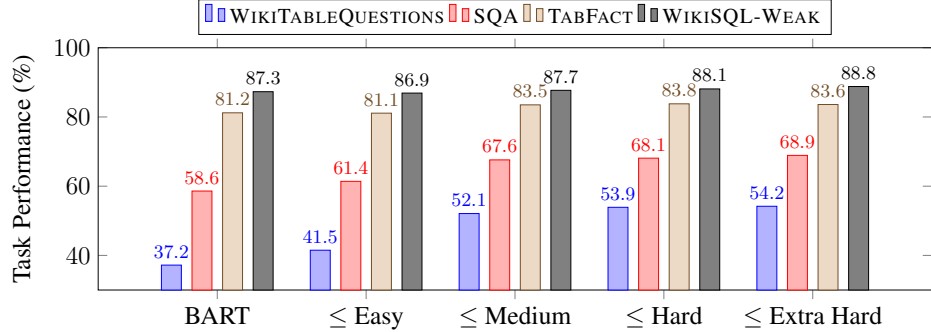

Figure 7: The performance of downstream tasks (dev sets) at different pre-training difficulties with the same amount of examples (0.5 Million). ≤ Medium means that we only use SQL query templates with a difficulty level less than or equal to Medium when synthesizing its pre-training corpus. Notably, ≤ Extra Hard is equivalent to using all SQL query templates.

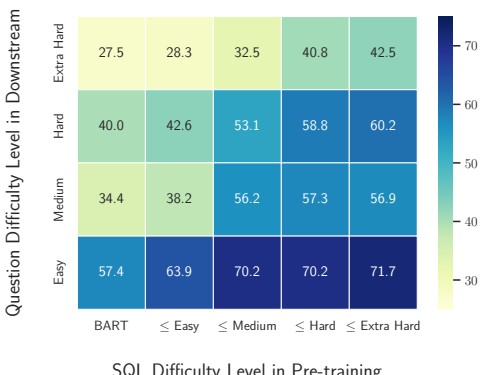

Figure 8: The fine-grained performance of different SQL difficulty levels in pre-training on different question difficulty levels from WIKITABLEQUESTIONS dev set.

**Fine-Grained Analysis** To understand the impact from a fine-grained perspective, we divide questions from the WIKITABLEQUESTIONS dev set into the same four levels of difficulty, with the help of SQL query annotation for WIKITABLEQUESTIONS questions provided by SQUALL. All fine-grained experimental results are presented in Figure 8. We can see that with the addition of harder SQL queries, the performance on questions at the same difficulty level are greatly improved. For example, the addition of Medium level SQL queries boosts the performance of Medium-level questions from 38.2% (≤ Easy) to 56.2% (≤ Medium), which is in line with expectations. More encouragingly, adding simpler SQL queries can even improve performance on harder questions. For example, compared to BART, the ≤ Medium pre-training leads to an impressive improvement of up to 13.1% in the performance of Hard-level questions.

## C.2 IMPACT OF NATURAL LANGUAGE IN PRE-TRAINING

**Natural Language Generation** Intuitively, compared to SQL queries, using NL sentences in pre-training is better for downstream tasks since the pre-training objective is nearly the same as the fine-tuning objective. However, it is non-trivial to obtain a fluent NL sentence which faithfully reflects the semantics of a SQL query. In this experiment, we follow Zhong et al. (2020a) to train a SQL-to-NL model and employ the model to translate SQL queries from the pre-training corpus into NL sentences. Concretely, our SQL-to-NL model is based on BART-Large (Lewis et al., 2020) and trained on the SQUALL dataset (Shi et al., 2020b), which contains nearly 9,000 SQL-NL pairs. Then we apply the well-trained SQL-to-NL model to the pre-training corpus of TAPEX (0.5 Million) and obtain a NL pre-training corpus of the same size. By manually analyzing 100 sampled translated NL sentences, we are surprised to find that all NL sentences are fluent, and nearly 68% of them

| SQL Query | Translated NL Sentence | Faithfulness |
|---|---|---|
| **SELECT** Name **WHERE** Age **>=** 28 | Who is at least 28 years old? | ✓ |
| **SELECT MAX** (Pick#) | What was the last pick in the 1989 major league baseball draft? | ✗ |
| **SELECT** Driver **ORDER BY** Pos **DESC LIMIT 1** | What driver came in last place? | ✓ |
| **SELECT COUNT** (Competition) **WHERE** Notes **!=** 100 | How many competitions have no notes? | ✗ |
| **SELECT COUNT** (*) **WHERE** Result = "won" **AND** Year **>** 1987 | How many times did they win after 1987? | ✓ |
| **SELECT MAX** (Chart Position) **−** **MIN** (Chart Position) **WHERE** Release date **=** "july 21, 1995" | What is the difference between the chart position of july 21, 1995 and the chart position of july 22, 1995? | ✗ |
| **SELECT** Nation **WHERE** Nation **!=** "Japan" **AND** Gold **=** (**SELECT** Gold **WHERE** Nation **=** "Japan" ) | Which other countries had the same number of gold medals as Japan? | ✓ |
| **SELECT** Incumbent Electoral History **GROUP BY** Incumbent Electoral History **ORDER BY COUNT** (*) **DESC LIMIT 1** | Who has held the office the most? | ✗ |

Table 10: The sampled SQL queries, their corresponding NL sentences translated by our SQL-to-NL model, and the faithfulness of the NL sentences.

| Setting | WIKISQL-WEAK | WIKITABLEQUESTIONS | SQA | TABFACT |
|---|---|---|---|---|
| TAPEX *with.* SQL | 88.8 | 54.2 | 68.9 | 83.6 |
| TAPEX *with.* NL | 87.5 | 52.8 | 68.7 | 83.7 |

Table 11: The downstream performance on dev sets of TAPEX with the SQL and the NL pre-training corpus. The NL corpus is obtained via translating the SQL corpus using our SQL-to-NL model, and they share the same amount of examples (0.5 Million).

are faithful to the semantics of the corresponding SQL queries. Table 10 presents some sampled SQL queries and their corresponding translated NL sentences. After obtaining the NL pre-training corpus, we follow the same pre-training and fine-tuning procedures as TAPEX to leverage it.

**Performance Comparison** We compare the performance of all downstream tasks between TAPEX *with.* SQL and TAPEX *with.* NL in Table 11. Surprisingly, the performance of TAPEX *with.* NL is comparable or even worse than the one of TAPEX *with.* SQL. For example, compared to using SQL queries in pre-training, using NL sentences causes a drop of $1.4\%$ on WIKITABLE-QUESTIONS. We attribute such drop to the fact that the translated NL sentences contain some noise. Taking the second row in Table 11 as an example, the translated NL sentence includes extra information such as "in the 1989 major league baseball draft", which may interfere with the pre-training.

# D FINE-GRAINED ANALYSIS OF SQL EXECUTION

Figure 9 provides a fine-grained analysis of the SQL execution accuracies for each operator type.

| Table | | | |
|---|---|---|---|
| Year | City | Country | Nations |
| 1896 | Athens | Greece | 14 |
| 1900 | Paris | France | 24 |
| 1904 | St. Louis | USA | 12 |
| 1908 | London | UK | 22 |
| … | … | … | … |
| 2004 | Athens | Greece | 201 |
| 2008 | Beijing | China | 204 |
| 2012 | London | UK | 204 |

| Execution Performance | | | |
|---|---|---|---|
| Operator | Example SQL | Percent | Accuracy |
| Select | SELECT City, Country | 100.0 | 89.6 |
| Filter | SELECT City WHERE Country = Greece | 72.4 | 90.6 |
| Aggregate | SELECT AVG (Nations) WHERE Year < 2000 | 34.2 | 89.9 |
| Superlative | SELECT City ORDER BY Year DESC LIMIT 1 | 31.2 | 89.3 |
| Arithmetic | SELECT MAX (Year) – MIN (Year) | 24.9 | 87.3 |
| Comparative | SELECT Country WHERE Year <= 2000 | 18.8 | 85.1 |
| Group | SELECT City GROUP BY City HAVING COUNT (*) > 1 | 4.3 | 84.2 |
| Sort | SELECT Country ORDER BY Year | 1.0 | 84.1 |
| Union & Intersection | SELECT City WHERE Country = Greece UNION SELECT City WHERE Country = USA | 0.3 | 89.4 |

Figure 9: The fine-grained statistics of typical operators, example SQLs, operator percentage and their execution accuracies on the held-out $20,000$ SQL queries.

