# OpenReview forum: "TAPEX: Table Pre-training via Learning a Neural SQL Executor"
_ICLR.cc/2022/Conference — ICLR 2022 Poster_

### Official Review · Reviewer_ukoF · 2021-10-27

**Correctness:** 4
**Technical Novelty And Significance:** 3
**Empirical Novelty And Significance:** 4
**Recommendation:** 8
**Confidence:** 4

**Details Of Ethics Concerns:**

Nan

**Main Review:**

Strengths
- the proposed pre-training strategy is simple and effective.
- the performance in the 4 evaluated benchmarks are significantly above the state-of-the-art.
- the paper provides many insights on the results (Figure 4&5&6, Table 6) and the paper is clearly written.

Weaknesses
- Table + Query vs text-to-SQL. This is not a weakness for this particular model but more for the general methodology of providing the entire table as input to the model. As the authors mention in the limitation section, the model has a limited input length, thus cannot handle big tables. This could be solved by generating SQL queries rather than learning how to execute them (which for a neural net is very hard), but yet again as mentioned by the authors in the limitation section, the TAPEX pre-training is not effective in this setting. I would not be surprised if the performance of the table as input + query setting is not as good as the semantic parsing one (e.g., text-to-SQL) in scenarios with very complex queries or very large tables. Anyhow, the proposed pre-training is very effective in this setting, so overall, this is not a major weakness in my view.
- Template for the synthetic generation. Although the performance with template-based data generation is good, I believe this is quite a restriction over the multitude of possible queries that can be generated using grammars.


**Summary Of The Paper:**

The authors propose a pre-training strategy for Table (structured data) grounded tasks (question answering and fact verification). The main idea is to teach the model to execute SQL queries, by predicting the execution of the query, rather than using a pre-training strategy over the tabular data (e.g., Masked LM pre-training, or masked column pre-training). To elaborate, the authors synthetically generate SQL queries, execute them over tables, and collect the execution output. Then, the model is trained to generate (in an autoregressive way) the execution output given the concatenation of the table and SQL query (Figure 3 for more details). In this way, the model is forced to reason over the table to generate an answer, rather than using a reconstruction loss (MLM) that improve the representation of the input.

Experiments
The authors use BART as the base model and use the SQL templates extracted in SQUALL (Shi et al. 2020) executed over 1500 randomly selected tables from WikiTable-Questions (Pasupat & Liang 2015) (from the training set and not overlapping with dev and test). The generated input-output pairs are 5M and the accuracy on a held-out is 89.6%. Then the pre-trained model is fine-tuned to 4 datasets (3 tables QA and 1 TableFact), and the results show a consistent and significant improvement on the existing state-of-the-art. Finally, the authors show an interesting analysis on the attention (Figure 4), the efficiency of the pre-trained task (Figure 5&6) and a manual evaluation on 500 samples from WikiTables showing how TAPEX improved over BART on different kinds of QA types (Table 6).


**Summary Of The Review:**

A simple and effective pre-training strategy for Table QA. The authors proposed TAPEX, a model trained on executing SQL query rather than MLM, a new SOTA in 4 benchmarks, and many analyses and visualizations.

---

> ### Author Response · Authors · 2021-11-23
> **Response to Reviewer ukoF**
>
> **W1. Table + Query vs Text-to-SQL.** Since TAPEX is in the scope of Table + Query, it does have some limitations as discussed in the Section-5-Analysis-Limitations. We will actively explore the application of TAPEX on the text-to-SQL task in our future work.
>
> **W2. Template for the synthetic generation.** Thanks for the valuable feedback! We agree that TAPEX would be more extensible if we use the SQL grammar to synthesize SQL queries in pre-training, and we will actively explore it in the future.

---

> ### Author Response · Authors · 2021-12-01
> **Response to Reviewer ukoF: Final Thanks**
>
> **We would like to thank you again for your detailed evaluation and constructive comments**. In particular, the two concerns you pointed out will be our important future work. We believe our approach will benefit the task of text-to-SQL in the near future, since our approach to some extent facilitates table understanding (Figure 4), which we think is also important for the task of text-to-SQL. We hope that this work would draw the community's attention to the promising direction of pre-training on synthetic data.
>
> Best,
>
> Paper1389 Authors

---

### Official Review · Reviewer_Subg · 2021-11-02

**Correctness:** 4
**Technical Novelty And Significance:** 3
**Empirical Novelty And Significance:** 3
**Recommendation:** 8
**Confidence:** 5

**Main Review:**

The paper is in general very easy to follow with a quite clear structure. The main technical part is quite simple yet effective. I believe simple is a good feature to have.

The proposed method is similar to the previous papers on "demonstrating transformers are universal approximator". The paper is motivated to  train the BART model to mimic the complex logical operations done by SQL query language. Though the accuracy is still not perfect, with "comparative/arithmetic" only achieving 50% accuracy, the other logical operations like group/select/filter are already pretty good.

I especially like the idea of "not using huge amount of mined noisy data for pre-training". As the huge amount of data can be very noisy and could waste large amount of time filtering and preprocessing. The paper demonstrates that small amount of tables are already quite good enough since the lexical/semantic variance is already captured during the BART pre-training, the model only needs to learn how to reason over the logical operations. This idea is very neat and proves effective.

There is some minor issues for the paper:
1. the paper mentions in multiple places that their model can address the problem of "how to design an effective pre-training task to exploit the table structure", I do have strong reservation for this point. First of all, the paper still uses the linearization to represent the table in the encoder side, no real structural information is leveraged in the model. The pre-training objective function is also irrelevant to the table structure. Therefore, this claim is flawed. I would prefer "how to design effective pre-training task to approximate the structural/arithmetic reasoning of formal language" would be a much better motivation.
2. Besides, the proposed algorithm is not specific to tables. I think KB can also be applied as long as there is formal language which can be synthesized. The gist of the paper is "mimicking logical operations in formal language" rather than "incorporating tables structures in pre-training". I would appreciate if the authors could revise their wording in the paper.

**Summary Of The Paper:**

This paper proposes a new approach to perform table-pretraining, this new approach is hugely different from the previous approaches like TaPas, TaBERT, etc in two ways: 1) the model architecture does not need to be designed to fit the table structure, no new objective functions have been invented, a simple BART model can be used to keep pre-training with its original encoder-decoder loss, 2) the algorithm does not require large amount of mined table-text parallel corpus, which avoids huge amount of data preprocessing and heuristics to do noise canceling.

The main technical part of this paper is "small amount of tables" while "large amount of synthesized SQL". The basic idea mimics the previous papers by (Yu et al. 2021) to utilize the clean dataset and templates to generate arbitrary amount of SQL queries and simulate various environments. The BART model is essentially used as a universal approximator to estimate the logical operations for SQL queries. The massively pre-trained TAPEX model indicates its superb SQL execution capability as indicated in Table 6, outperforming the BART without such SQL-approximation pre-training by a huge margin of 20%+. This is quite impressive, and demonstrates the strong approximation capability of BART model even without specially-designed logic-units.

The empirical results are also very impressive, the model matches or surpasses the previous SoTA on all the four datasets, which is definitely a plus for the proposed model.

**Summary Of The Review:**

To sum up, the paper has strong empirical evidence to support its simple yet effective approach. Though the wording and claims are sometimes not quite accurate, it's still a very good paper. I believe it can arouse lots of interest in the community and persuade more people to abandon "mined large noisy corpus" and turn to "accurate and small corpus". Based on these contributions, I would recommend accepting this paper.

---

> ### Author Response · Authors · 2021-11-23
> **Response to Reviewer Subg**
>
> **W1 & W2 The wording of the paper.**
>
> Thanks for the constructive feedback on the paper wording! We agree that the claim of "exploit the table structure" is flawed and we should highlight "mimicking logical operations in formal language" instead of "table structure".
>
> Therefore, we have carefully revised the wording of the paper following your suggestions. The major changes include:
> - *The contribution part in the abstract.* We revise the original contribution from "TAPEX is the first one to exploit structured tabular data via pre-training on synthetic executable programs" to "TAPEX is the first one to explore table pre-training via synthetic executable programs".
> - *The challenge part in the introduction (Paragraph 2).* We modify the second challenge from "how to design an effective pre-training task to exploit the tabular structure" into "how to design an efficient pre-training task for table pre-training". Meanwhile, in summarizing the shortcomings of previous works, we remove the description "the inefficient exploitation of tabular structures".
> - *The motivation part in the introduction (Paragraph 3).* We revise the motivation of TAPEX from "achieves efficient table pre-training via exploiting the executability of tables" to "achieves efficient table pre-training via approximating the structural reasoning process of formal languages over tables".
>
> We hope these revisions will make our paper clearer, and please kindly find more details on the latest draft.

---

> > ### Comment · Reviewer_Subg · 2021-12-01
> > **Thanks for update**
> >
> > I have read the rebuttal and am mostly satisfied withe updates being made. I would keep my score to recommend an acceptance.

---

> > > ### Author Response · Authors · 2021-12-01
> > > **Response to Reviewer Subg: Final Thanks**
> > >
> > > We are very happy to see that you are mostly satisfied with our revisions. **We would like to thank you again for the constructive comments, which have guided us to better present our work**. We are also very grateful for your excellent summary:
> > > > I believe it can arouse lots of interest in the community and persuade more people to abandon "mined large noisy corpus" and turn to "accurate and small corpus"
> > >
> > > That motivates us a lot and makes us more confident on the direction of using synthetic and accurate data for pre-training.
> > > Last, your proposed future work on KB is also interesting and exciting. We believe that our work took the first step and hope that this work would draw the community's attention to the promising direction of pre-training on executable programs.
> > >
> > > Best,
> > >
> > > Paper1389 Authors

---

### Official Review · Reviewer_xsFm · 2021-11-03

**Correctness:** 4
**Technical Novelty And Significance:** 4
**Empirical Novelty And Significance:** 3
**Recommendation:** 8
**Confidence:** 4

**Details Of Ethics Concerns:**

N/A.

**Main Review:**

Strengths:
1. The paper is very clear and informative, well-written with abundant figures & tables for better understanding. The introduction to pretraining / fine-tuning methods is well arranged. Thanks for the effort!
2. The idea of learning a SQL executor for pretraining is simple yet effective. It’s kinda novel to use it as a way of pretraining and the authors conducted many experiments to demonstrate its effectiveness.
3. The authors conducted many ablation studies about how much the volume of pretraining data impacts the final result, where does the performance gain come from and how does the dataset impact the results. The experiments further confirmed the effectiveness of the method.
4. The model and datasets proposed in this paper are publicly available, making it easy to reproduce.
5. The results demonstrate a significant improvement over the previous works.

Weaknesses:
1. The paper has a kinda novel idea on the pretraining method but the problem setting is limited to table understanding, limiting the method to only table related problems.
2. The paper experimented using SQL style query language and the examples seemed simple. While TableQA is a challenging problem, some analysis on the impact of difficulty of queries on the final performance would be appreciated.
3. The claim that the model can reproduce the table structure seems to be a bit questionable. With the attention mechanism, the attention weights are very commonly calculated and shown as in Figure 4. It would be supported if the weights can be accumulated on some rows or columns.
4. While SQL language is structured, natural language questions can be very flexible. Would appreciate it if there are more comparisons on the change of performance if we generate the SQL in a different way or use paraphrase of question.


**Summary Of The Paper:**

This paper proposed `TAPEX`, a table pretraining method that conducts pretraining via learning a neural SQL executor over a synthetic corpus. Such corpus is obtained by automatically synthesizing executable SQL queries and execution results. By selecting a diverse, large scale and high quality synthetic corpus, the method claimed to address the data scarcity challenge. Then the executor could be further fine-tuned for downstream tasks, and demonstrate significant improvements over previous state-of-the-art work on table question answering and fact verification datasets.

**Summary Of The Review:**

This paper is well-written and very informative, introducing `TAPEX`, a pretraining method learning SQL executor, with significant performance improvement. The novel design of pretraining is very extensive and proven to be effective on TableQA setting. While some more experiments could be conducted to further understand the performance gain, the current paper has done a lot of ablation study, showing the impact of pretraining data. I recommend this paper to be accepted.

---

> ### Author Response · Authors · 2021-11-23
> **Response to Reviewer xsFm**
>
> **W1. The method is limited to table related problems.** Yes, our current paper mainly focuses on table related tasks. It is indeed valuable to explore the potential of TAPEX for other natural language tasks, and this will be an important piece of our future work.
>
> **W2. Analysis on the impact of SQL query difficulty.** During the rebuttal phase, we have performed two kinds of analysis on the impact of difficulty of SQL queries. First, we analyze the impact across different downstream tasks. The experimental results demonstrate that adding harder SQL queries to the pre-training corpus always contributes to the downstream performance. Second, we analyze the impact in a fine-grained perspective on WikiTableQuestions. The fine-grained experimental results show that simple SQL queries in pre-training can surprisingly boost the performance on hard questions. Please kindly find details in Appendix C.1 of the latest draft.
>
> **W3. The claim that the model can reproduce the table structure seems to be a bit questionable.** Thanks for the feedback! We have revised the wording of the paper from "TAPEX seems to restore the structure of flattened tables internally" to "TAPEX seems to focus more on the row and the header where a cell corresponds to". Hope the revision alleviates your concern.
>
> **W4. Analysis on the natural language in pre-training.** During the rebuttal phase, we have performed the analysis on the impact of natural language on downstream tasks. Concretely, we train a SQL-to-NL model based on BART-Large and employ the model to translate SQL queries from the pre-training corpus into NL sentences. After pre-training on the NL pre-training corpus, we compare the performance between TAPEX with NL and TAPEX with SQL. We are surprised to find that the performance of TAPEX with NL is comparable or even worse than the one of TAPEX with SQL. Please kindly find details in Appendix C.2 of the latest draft.

---

> > ### Comment · Reviewer_xsFm · 2021-11-23
> > **Response to Authors' Rebuttal**
> >
> > Thanks to the authors for the very detailed update in response to my review. The response has addressed most of my concerns and I am more confident to recommend this paper to be accepted.

---

> > > ### Author Response · Authors · 2021-12-01
> > > **Response to Reviewer xsFm: Final Thanks**
> > >
> > > We are very happy to see that most of your concerns are addressed. **We would like to thank you again for the precious comments and also for increasing the confidence on accepting our paper**. The suggested analysis and experiments brought more insights for us to understand the approach itself. We believe that our work took the first step and hope that this work would draw the community's attention to the promising direction of pre-training on executable programs beyond table-related tasks.
> > >
> > > Best,
> > >
> > > Paper1389 Authors

---

### Official Review · Reviewer_YHKE · 2021-11-30

**Correctness:** 1
**Technical Novelty And Significance:** 3
**Empirical Novelty And Significance:** 3
**Recommendation:** 6
**Confidence:** 3

**Main Review:**

Strength:
- Formulating table pre-training through learning via synthetic SQL corpus is a novel idea.
- The proposed new pretraining paradigm of table pretraining is simple while effective. Moreover, the authors show its efficiency on a much smaller corpus.

Weakness:
- Compared with naming it as 'pre-training', the proposed approach should be named as data augmentation from constructed synthetic table corpus. It is intuitive to see the improvement of performance since the input and output format between synthetic corpus and the downstream task (QA, fact verification) are the same. Therefore, there is no wonder a huge improvement is received. This work is less interesting and novel from this perspective.

**Summary Of The Paper:**

This paper proposes a new table pre-training approach. Different from previous masked language approaches for table pre-training. The authors propose to pre-train (or train) BART on the synthetic SQL corpus. The input for pretraining is the concatenation of SQL query and table while the output is the results by executing the SQL query through the SQL execution engine. Results show that the proposed pretraining approach reduces the gap between pre-training and fine-tuning on table understanding tasks including table-based question answering and table-based fact verification -- it outperforms BART and baselines on four datasets.

**Summary Of The Review:**

I lean to accept this work since it provides a new view of pretraining and the proposed approach receives significant improvement on four datasets.

---

> ### Author Response · Authors · 2021-11-30
> **Response to Reviewer YHKE [2/2]**
>
> ### W2. It is intuitive to see the improvement of performance since the input and output format between synthetic corpus and the downstream task (QA, fact verification) are the same.
>
> We must say that we cannot agree with that point because:
>
> 1. **SQL queries are fundamentally different from NL sentences, and thus the input of our synthetic corpus is not "the same" as the one of downstream task**.
>
>     To the best of our knowledge, none of previous works has explored using non-natural-language (e.g., programming language such as SQL) during pre-training to boost the performance of natural language related downstream tasks. People might think pre-training on SQL queries is hardly beneficial for downstream tasks, because SQL and NL are fundamentally different. For example, SQL queries follow a set of syntax rules, which makes them easy to synthesize, but this is not the case for NL sentences. Considering this fundamental difference, we disagree with the argument that the input format is the same between the synthetic corpus and the downstream task.
>
> 2. **The downstream task Table-based Fact Verification (TableFV) is a sequence classification task, whose output is not "the same" as our synthetic corpus (i.e., SQL Execution), a sequence generation task**.
>
>     However, our pre-training also boosts the performance of TableFV by a large margin. As stated in the paper,
>
>     > For TableFV, as BART does for sequence classification tasks (Lewis et al., 2020), the same input is fed into both the encoder and decoder, and a binary classifier upon the hidden state of the last token in the decoder is used for the output.
> the output of TableFV is obtained by passing the last decoder hidden state into a binary classifier which is not pre-trained during the pre-training.
>
>     As can be seen, the output format between the pre-training task and TableFV is also totally different. Therefore, we disagree with the argument that the output format is the same between the synthetic corpus and the downstream task.
>
>
> Best,
>
> Paper1389 Authors

---

> ### Author Response · Authors · 2021-11-30
> **Response to Reviewer YHKE [1/2]**
>
> Thank you for the comment, but we must say that we cannot fully agree with the comment. Below are detailed responses for each weakness mentioned in your comment.
>
> ### W1. The naming of the proposed approach
>
> We present below three reasons of why our approach is better suited to be called pre-training instead of data augmentation:
>
> 1. **Our approach is a pre-training method since it well aligns with the classical pre-training paradigm**
>
>     In the research community, it is common sense to use the term pre-training when an approach first trains on a collected large-scale corpus and then fine-tunes on a downstream task. Our work follows the same pre-training-fine-tuning paradigm, so we formulate our approach as a pre-training approach.
>
> 2. **Our approach should be a pre-training method since similar works claim their methods as pre-training methods**
>
>     Pre-training on structured tabular data is an active research area in recent years, which has been explored by a number of excellent previous works [1,2,3,4,5,6]. Similar to our work, they first train their models on a collected table-related corpus in their methods. To the best of our knowledge, all of these works refer to their methods as pre-training methods.
>
> 3. **Our approach is fundamentally different from data augmentation methods**
>
>     As stated by previous studies [7], data augmentation methods usually require ensuring that the augmented data is in the same distribution of the original data. However, our corpus consists of synthetic SQL queries, which are fundamentally different from natural language sentences of downstream tasks. The fundamental difference is contradictory with the principle of data augmentation, suggesting that our approach is not a data augmentation method.
>
> In summary, our approach follows the classical pre-training-fine-tuning paradigm, with a similar pre-training process to previous table pre-training works, and is fundamentally different from data augmentation methods. Therefore, we believe that the naming pre-training is better than data augmentation for our approach.
>
> ### Reference
>
> [1]. Jonathan Herzig, Pawel Krzysztof Nowak, Thomas M ̈uller, Francesco Piccinno, and Julian Eisen-schlos. TAPAS: Weakly supervised table parsing via pre-training. https://arxiv.org/abs/2004.02349
>
> [2]. Xiang Deng, Ahmed Hassan Awadallah, Christopher Meek, Oleksandr Polozov, Huan Sun, and Matthew Richardson. Structure-grounded pretraining for text-to-SQL. https://arxiv.org/abs/2010.12773
>
> [3]. Julian Eisenschlos, Syrine Krichene, and Thomas Muller. Understanding tables with intermediate pre-training. https://arxiv.org/abs/2010.00571
>
> [4]. Pengcheng Yin, Graham Neubig, Wen-tau Yih, and Sebastian Riedel. TaBERT: Pretraining for joint understanding of textual and tabular data. https://arxiv.org/abs/2005.08314
>
> [5]. Tao Yu, Chien-Sheng Wu, Xi Victoria Lin, Bailin Wang, Y. Tan, Xinyi Yang, Dragomir Radev,R. Socher, and Caiming Xiong. Grappa: Grammar-augmented pre-training for table semantic parsing. https://arxiv.org/abs/2009.13845
>
> [6]. Tao Yu, Rui Zhang, Alex Polozov, Christopher Meek, and Ahmed Hassan Awadallah. Score: Pre-training for context representation in conversational semantic parsing. https://openreview.net/forum?id=oyZxhRI2RiE
>
> [7]. Bohan Li, Yutai Hou, and Wanxiang Che. Data Augmentation Approaches in Natural Language Processing: A Survey. https://arxiv.org/abs/2110.01852

---

> ### Author Response · Authors · 2021-12-01
> **Gentle Reminder to Reviewer YHKE: Any further question?**
>
> Thank you again for your comment.
>
> Since the discussion period is coming to an end, we would like to know if you have any further questions. From our point of view, we feel that we have clarified all the concerns and misunderstandings that you have raised in the initial review.
>
> **If you could re-assess our paper based on our response and let us know if you might find our work more positive**, we would be very grateful! If you have any further questions, please ask them and we are available at any time for further clarification.
>
> Best,
>
> Paper1389 Authors

---

> > ### Comment · Reviewer_YHKE · 2021-12-02
> > **Thanks for your response**
> >
> > Thanks. I agree with your response on W1 while keeping my view on W2.

---

> > > ### Author Response · Authors · 2021-12-02
> > > **Response to Reviewer YHKE**
> > >
> > > Thanks for your comment. We would like to clafiy that our response W2 is mainly for the following statement from your initial review:
> > >
> > > > It is intuitive to see the improvement of performance since the input and output format between synthetic corpus and the downstream task (QA, fact verification) are the same.
> > >
> > > We have to emphasize that we rellay disagree the statement because:
> > >
> > > - Having clear intuition behind an algorithm design is a common practice, and it is also important to provide insights for future work. Given this, it cannot be a flaw for a method. Therefore, we cannot understand why the intuitive effectiveness of our approach (up to 19% absolute improvement) can be a weakness. Could you kindly elaborate it in detail?
> > > - We think the statement "since the input and output format ... are the same" is scientifically wrong. To emphasize again, the synthetic corpus consists of SQL queries, while downstream tasks consist of NL sentences. Based on this fact, your statement is equivalent to saying that NL sentences and SQL queries are "the same". We suspect that other researchers would not agree with your claim that programming languages and natural languages can be "the same".
> > >
> > > We sincerely hope you can re-assess our paper based on above two points.
> > >
> > > With the highest respect,
> > >
> > > Paper1389 Authors

---

> > > > ### Comment · Reviewer_YHKE · 2021-12-02
> > > > **Response**
> > > >
> > > > I DO acknowledge the difference between SQL queries and NL questions. Please do not wrongly explain my comments.
> > > >
> > > > This work inherits the idea of [1] through further pretraining BART with data from a similar domain. Compared with directly finetuning BART (a model pretrained under denoising target) to the downstream table understanding tasks, the gap between pretraining and finetuning is narrowed because you construct synthetic table corpus with SQL queries which aligns the finetuning tasks (table understanding tasks with NL questions) better.
> > > >
> > > > I do appreciate the contribution to the proposed pretraining task and the performance improvement. Please be polite before making any response.
> > > >
> > > > [1] Don’t Stop Pretraining: Adapt Language Models to Domains and Tasks

---

> > > > > ### Author Response · Authors · 2021-12-02
> > > > > **Response to Reviewer YHKE: Final Thanks**
> > > > >
> > > > > First of all, we would like to thank you for your quick response and active discussion, especially as we are getting closer to the end of the discussion stage.
> > > > >
> > > > > As you stated, the line of table pre-training [1,2,3,4,5,6], including our paper, inherit the idea of [7] through further pre-training language models with data from a similar domain. Compared to previous works using natural language sentences for pre-training, our work is the first one to show that pre-training with easily collected programs can be powerful, which is also the main contribution of our work. We hope that this work would draw the community's attention to the promising direction of pre-training with programs.
> > > > >
> > > > > Finally, although our view differs from yours, we greatly appreciate the time and effort you spent in reviewing this paper. Thank you very much!
> > > > >
> > > > > Best,
> > > > >
> > > > > Paper1389 Authors
> > > > >
> > > > > ### Reference
> > > > >
> > > > > [1]. Jonathan Herzig, Pawel Krzysztof Nowak, Thomas M ̈uller, Francesco Piccinno, and Julian Eisen-schlos. TAPAS: Weakly supervised table parsing via pre-training. https://arxiv.org/abs/2004.02349
> > > > >
> > > > > [2]. Xiang Deng, Ahmed Hassan Awadallah, Christopher Meek, Oleksandr Polozov, Huan Sun, and Matthew Richardson. Structure-grounded pretraining for text-to-SQL. https://arxiv.org/abs/2010.12773
> > > > >
> > > > > [3]. Julian Eisenschlos, Syrine Krichene, and Thomas Muller. Understanding tables with intermediate pre-training. https://arxiv.org/abs/2010.00571
> > > > >
> > > > > [4]. Pengcheng Yin, Graham Neubig, Wen-tau Yih, and Sebastian Riedel. TaBERT: Pretraining for joint understanding of textual and tabular data. https://arxiv.org/abs/2005.08314
> > > > >
> > > > > [5]. Tao Yu, Chien-Sheng Wu, Xi Victoria Lin, Bailin Wang, Y. Tan, Xinyi Yang, Dragomir Radev,R. Socher, and Caiming Xiong. Grappa: Grammar-augmented pre-training for table semantic parsing. https://arxiv.org/abs/2009.13845
> > > > >
> > > > > [6]. Tao Yu, Rui Zhang, Alex Polozov, Christopher Meek, and Ahmed Hassan Awadallah. Score: Pre-training for context representation in conversational semantic parsing. https://openreview.net/forum?id=oyZxhRI2RiE
> > > > >
> > > > > [7]. Suchin Gururangan, Ana Marasović, Swabha Swayamdipta, Kyle Lo, Iz Beltagy, Doug Downey and Noah A. Smith. Don't Stop Pretraining: Adapt Language Models to Domains and Tasks. https://arxiv.org/abs/2004.10964v1

---

### Author Response · Authors · 2021-11-23
**General Response: Summary of the Update**

We really appreciate all reviewers for their careful reviews and constructive comments. We have submitted an **updated version** of the paper based on these comments and included more experimental results. We hope that the latest draft addresses your questions.

- We add an exploratory analysis of pre-training, including the impact of SQL query difficulty on downstream performance (Appendix C.1) and the comparison between SQL pre-training and natural language pre-training (Appendix C.2) suggested by Reviewer `xsFm`.
- We revise the wording of the paper according to comments from Reviewer `xsFm` and Reviewer `Subg`.
- We respond to all weaknesses raised by Reviewer `xsFm`, Reviewer `Subg`, and Reviewer `ukoF`. We hope these responses alleviate the concerns of reviewers.

---

### Decision · Program_Chairs · 2022-01-20

**Decision:**

Accept (Poster)

**Comment:**

Reviewers are positive overall -- the is a general consensus towards acceptance. Reviewers viewed the simplicity, novelty, and effectiveness of the propose pre-training approach as strengths. Further, reviewers praised the draft as very clearly written, and viewed experimental ablations as relatively in-depth -- e.g. two reviewers found the additional analysis of impact of data size to be valuable. A few concerns about additional ablations and claims were brought up, but all were adequately addressed in author response.